# NOD-like Receptor Signaling Pathway in Gastrointestinal Inflammatory Diseases and Cancers

**DOI:** 10.3390/ijms241914511

**Published:** 2023-09-25

**Authors:** Yujie Zhou, Songyan Yu, Wenyong Zhang

**Affiliations:** 1School of Medicine, Southern University of Science and Technology, Shenzhen 518055, China; 12110522@mail.sustech.edu.cn (Y.Z.); 12112023@mail.sustech.edu.cn (S.Y.); 2Key University Laboratory of Metabolism and Health of Guangdong, Southern University of Science and Technology, Shenzhen 518055, China

**Keywords:** NOD-like receptor, gastrointestinal inflammatory diseases, esophageal cancer, gastric cancer, colorectal cancer

## Abstract

Nucleotide-binding and oligomerization domain (NOD)-like receptors (NLRs) are intracellular proteins with a central role in innate and adaptive immunity. As a member of pattern recognition receptors (PRRs), NLRs sense specific pathogen-associated molecular patterns, trigger numerous signaling pathways and lead to the secretion of various cytokines. In recent years, cumulative studies have revealed the significant impacts of NLRs in gastrointestinal (GI) inflammatory diseases and cancers. Deciphering the role and molecular mechanism of the NLR signaling pathways may provide new opportunities for the development of therapeutic strategies related to GI inflammatory diseases and GI cancers. This review presents the structures and signaling pathways of NLRs, summarizes the recent advances regarding NLR signaling in GI inflammatory diseases and GI cancers and describes comprehensive therapeutic strategies based on this signaling pathway.

## 1. Introduction

The immune system consists of the innate immune system and the adaptive immune system. To recognize endogenous pathogen-associated molecular patterns (PAMPs) or external damage-associated molecular patterns (DAMPs) in the innate immune system—which is the first line of defense to fight infection—a special kind of molecules called pattern recognition receptors (PRRs) have developed. PRRs can first bind with a series of common structures on pathogens and apoptotic or damaged cells and then trigger the initiation and execution of immune responses via various signaling pathways [1]. PRRs are divided into two major categories based on their cellular locations; toll-like receptors (TLRs) and C-type lectin receptors (CLRs) are transmembrane proteins, while retinoic acid-inducible gene (RIG)-I-like receptors (RLRs) and NOD-like receptors (NLRs) are cytoplasmic [2]. In addition to NLRs’ originally discovered function of responding to endogenous byproducts of tissue damage or intracellular pathogens, many recent studies have shown that NLRs play important roles in cell death, inflammatory reactions and even tumorigenesis. There has been evidence that NLRs can be expressed in adaptive immune cells and participate in the adaptive immune system [3].

Gastrointestinal (GI) inflammatory diseases and GI cancers are serious health problems. The incidence and mortality of GI inflammatory diseases and GI cancers remain high and are even on the rise. For example, the prevalence of inflammatory bowel diseases (IBD) in Western countries has risen to 0.5% of the general population [4]. As the fourth most common cause of cancer-related death, gastric cancer has a high disease burden, with over 1 million new cases annually [5,6]. The pathogenesis of GI inflammatory diseases and GI cancers is intricate and needs more research to elucidate it. The usual mechanisms of GI inflammatory diseases and GI cancers involve host genetics, environmental risk factors, an imbalance of gut microbiota and immune dysregulation. Recently, cumulative evidence has shown that NLRs play a significant role in GI inflammatory diseases and GI cancers, including gastritis, IBD, celiac disease, esophagus cancer, gastric cancer and colorectal cancer. In this review, we summarize the recent progress on the NLR signaling pathways, especially in GI inflammatory diseases and GI cancers, and discuss the potential treatment strategies based on the current understanding of the pathogenic roles of NLRs.

## 2. The NLR Family and Correlative Signaling Pathways

NLRs were first discovered in plants in the 1990s. Initially, these proteins were named nucleotide binding-leucine rich repeats (NB-LRRs). NB-LRRs were then discovered to function in recognizing antigens and activating the immune response discovered in vertebrate species, including humans [7]. As a significant group of PRRs, NB-LRRs in humans—known as NLRs—recognize PAMPs or DAMPs in the cytosol. NLRs typically consist of three domains, including a N-terminal regulatory domain, a nucleotide-oligomerization-binding domain (NOD)—also known as the NACHT domain—and a C-terminal leucine-rich repeat domain (LRR) (Figure 1). The combination of five different functional elements in the N-terminal domain—namely the acidic transactivation domain (AD), pyrin domain (PYD), caspase recruitment domain (CARD), death effector domain (DED) and baculovirus inhibitor repeat domain (BIR)—classify NLRs into five subfamily members: NLRA, NLRB, NLRC, NLRP and NLRX (Figure 1). Both the NACHT domain and the LRR are conservative and present in almost all recognized NLRs, with the exception of NLRP10 [8]. The NACHT domain is named after neuronal apoptosis inhibitor protein (NAIP), major histocompatibility complex (MHC) class II transcription activator (CIITA), incompatibility locus protein from *Podospora anserina* (HET-E) and telomerase-associated protein 1 (TP1). The NACHT domain transfers the signal to the N-terminal domain [9]. The LRR domain recognizes and binds with PAMPs and DAMPs to activate the NACHT domain.

### 2.1. NLRA

NLRA, also known as CIITA, is a class of NLRs characterized by an N-terminal acidic transactivation domain and a region rich in prolines, serines and threonines (P/S/T domain) [10]. As the master transcriptional regulator of MHC class II (MHC II), CIITA shuttles between the cytoplasm and nucleus [11]. The activity of CIITA is dependent on the self-association and oligomerization induced by the GTP binding domain and LRR domain [12,13,14,15]. CIITA is considered to have dual roles in regulating MHC gene transcription, even though there is no clear evidence that it binds directly to DNA [10]. Studies have shown that CIITA regulates the transcription of MHC-II genes by recruiting general transcriptional factors (e.g., TFIID, TFIIB) and chromatin remodeling coactivators (e.g., P300, CBP) and by its TAF1-like properties, such as acetyltransferase (AT) and kinase activities [16,17,18,19].

### 2.2. NLRB

The NLRB subfamily in humans comprises only one member: NAIP. NAIP contains a significant N-terminal BIR domain, which is thought to be connected with inflammasome formation [20]. NAIP in the cytoplasm recognizes and binds to bacterial flagellin and components from bacterial type III secretion systems (T3SS), including T3SS rod and needle proteins [21,22,23]. Then, this NAIP complex nucleates the assembly of the NLRC4 inflammasome, which further activates caspase 1 (CASP1) to cause the release of pro-inflammatory mediators [24]. In addition, according to some studies, NAIP inhibits the apoptosis induced by varieties of signals via the inactivation of CASP3, CASP7 and CASP9 [25].

### 2.3. NLRC

NLRC, the second largest subfamily of NLRs, is classified by the existence of N-terminal oligomerization CARD. NLRC consists of five members; namely, nucleotide oligomerization domain 1 (NOD1, NLRC1), nucleotide oligomerization domain 2 (NOD2, NLRC2), NLRC3, NLRC4 and NLRC5. For NLRC3 and NLRC5, due to possessing similar characteristics to other NLRCs in their homology and phylogenetic relationship, they are grouped into this subfamily, particularly with unknown N-terminal domains [26].

NOD1 and NOD2 are the primary members of NLRCs, with the N-terminus consisting of one CARD (for NOD1) or two CARDs (for NOD2). These two NOD proteins sense the peptidoglycan (PGN) from bacteria, including γ-D-glutamyl-meso-diaminopimelic acid (iE-DAP) for NOD1 and muramyl dipeptide (MDP) for NOD2 [27]. After binding to PGN at LRRs, the NACHT domain self-oligomerizes and is followed by CARD-CARD interaction [28]. Then, oligomerized NOD proteins with a scaffold from the endosomal membrane interact with the serine/threonine receptor-interacting protein 2 (RIP2) kinase, which further activates nuclear factor κB (NF-κB) via the ubiquitination of the NF-κB essential modulator (NEMO)/IKKγ complex and mitogen-activated protein kinase (MAPK) via the mediation of TAK1-associated binding protein (TAB) and its associated kinase TGF-β-activated kinase 1 (TAK1) to produce inflammatory cytokines [26,27,29]. In addition, several signaling pathways, including the NOD1/TNF receptor-associated factor 3 (TRAF3) pathway, NOD2/mitochondrial antiviral signaling (MAVS) protein pathway and NOD proteins/autophagy-related proteins (ATG) pathway, can lead to host defense and autophagic degradation [26,30].

NLRC3, a member of the NLRC subfamily discovered in 2005, is also known as CLR16.2 or NOD3 [31]. As a negative regulator, NLRC3 inhibits the NF-κB signaling pathway through decreasing the K63-linked polyubiquitination of TRAF6 or increasing the K48-linked polyubiquitination of interleukin-1 receptor-associated kinase1 (IRAK1) to impact both the innate immunity and adaptive immunity [32,33]. In addition, under the simulation from cytosolic DNA, cyclic di-GMP and DNA viruses, NLRC3 is a barrier to the interaction between the stimulator of interferon gene (STING) and TANK-binding kinase (TBK1) to prevent the production of type I interferon [34]. Recently, some studies have revealed that NLRC3 prevents inflammasome formation via interacting with apoptosis-associated speck-like protein (ASC) and pro-CASP1 and suppresses the PI3K-AKT-mTOR signaling pathway [35,36].

Containing one CARD at the N-terminus, NLRC4 mainly interacts with NAIP to form inflammasomes. Because of acidic residues on the NACHT domain in NLRC4, every NAIP with basic residues interacts with about 10 NLRC4 protomers to form an inflammasome [37]. Following the oligomerization of the CARD domain, the pro-forms of Gasdermin D (GSDMD), cytokines Interleukin- (IL-) 1β and IL-18 are cleaved successively after the activation of CASP1, leading to cytokine release and cell pyroptosis [38].

NLRC5, a special member in the NLRC subfamily with an atypical CARD lacking an acidic domain, is similar to CIITA. In contrast to CIITA, NLRC5 mainly upregulates MHC I and correlative proteins [39]. NLRC5 has been shown to suppress the NF-κB pathway through inhibiting the phosphorylation and kinase activity of IKKα/β [40].

### 2.4. NLRP

In humans, the NLRP is the largest subfamily of the NLRs, with a total of 14 members currently reported. It is characterized by an N-terminal PYD domain that can recruit inflammasome-activating scaffold protein ASC through homologous interactions to control the activation of CASP1. Once CASP1 is activated, pro-IL-1β and pro-IL-18 will be cleaved to form mature IL-1β and IL-18, and GSDMD will be produced to induce the pyroptosis of cells, thus affecting the innate and adaptive immunity [41,42,43].

NLRP1 is the first known PRR associated with inflammatory body formation. The difference is that the NLRP1 has a C-terminal extension containing a function-to-find domain (FIIND) and a CARD. Proteolysis in FIIND is essential for the activation of NLRP1 inflammasome [44]. The C-terminal CARD domain is responsible for recruiting ASC and promoting the synthesis of inflammatory bodies.

NLRP3 is the most extensively studied, and its abnormal activation is associated with a variety of inflammatory diseases. At present, a two-signal model is proposed for NLRP3 activation. Activation signal 1 is provided by microorganisms and endogenous cytokines, and activation signal 2 is provided by extracellular ATP, pore-forming toxins, etc. [45].

There is also considerable research on NLRP6 and NLRP12. NLRP6 is involved in the negative regulation of NF-κB and MAPK signaling pathways [46,47,48]. It is believed to be closely related to IBD and gastrointestinal cancer [49]. NLRP12 also plays an important role in the inhibition of NF-κB and MAPK signaling pathways [50]. It affects immune cell localization and recruitment, negatively regulates type I interferon (IFN-I) and inhibits tumor necrosis factor-α (TNF-α) [51,52,53].

Other members of the NLRP subfamily have also come under increasing scrutiny in recent years. NLRP2 is believed to play an important role in embryonic development, and previous studies have shown that it can also inhibit the activation of the NF-κB signaling pathway. Studies found that the role of NLRP2 in influencing the NF-κB signaling pathway may be twofold, which is related to its different functions in different cells [54,55,56]. NLRP7 has recently been found to be involved in the differentiation of decidual macrophages. In *in vitro* models, NLRP7 is up-regulated in both M1 macrophages and M2 macrophages after induction, and the overexpression of NLRP7 can inhibit the production of M1 macrophage cytokine and enhance the production of M2 macrophage cytokine; however, the specific mechanism remains to be further studied [57]. NLRP10 is the only NLR protein that lacks an NRR domain. Recent studies have revealed that NLRP10 can monitor mitochondrial integrity in a way that is independent of mitochondrial DNA, providing new targets for the understanding and treatment of mitochondrial diseases [58]. NLRP10 has also been found to play an important role in the development of inflammatory bowel disease in a model using mouse colonic epithelial cells [59].

### 2.5. NLRX

NLRX1 is the only reported member of the NLRX subfamily and is the first NLR protein to be shown to exist in mitochondria [60]. In addition to its role in regulating innate immunity and inflammation, NLRX also affects mitochondrial ROS production and is involved in the regulation of the NF-κB and JNK signaling pathways [61].

## 3. NLRs in GI Inflammatory Diseases

The most prevalent diseases in the GI tract are inflammatory diseases, including various forms of gastritis, enteritis and colitis. Among the possible mechanisms of GI inflammatory diseases, the abnormal regulation of the immune responses plays a significant role in their pathogenesis. As canonical NLR signaling pathways sense antigens and stimulate immune responses, their potential roles in representative GI inflammatory diseases are discussed below.

### 3.1. Gastritis

Gastritis is the inflammation of the gastric mucosa, which can be divided into acute gastritis and chronic gastritis. *Helicobacter pylori* (*H. pylori*)-associated gastritis is the main cause of chronic gastritis. In China, the overall infection rate of *H. pylori* is close to 50%. After infection, *H. pylori* interacts with the target cells in the gastric mucosa, activating multiple innate immune signaling pathways [62]. Extensive studies have revealed that the NLR signaling pathways are involved in *H. pylori*-associated gastritis. Gastric inflammatory responses are largely induced by the secretion of virulence factors of *H. pylori* through the type IV secretion system (T4SS) [63]. As mentioned above, two members in the NLR family, NOD1 and NOD2, can be activated by peptidoglycan from a broad range of bacteria. In response to the infection, NOD2 can mediate the transcription of immune response genes through the NF-κB pathway. According to study by El-Omar et al., the major cytokine secreted after *H. pylori* infection is IL-1β [64]. Yamauchi et al. found that the interruption of NF-κB signaling impacts pro-IL-1β and pro-IL-18 production in *H. pylori*-infected epithelial cells and macrophages [65]. Taken together, NOD2 appears to have the ability to recognize *H. pylori*, which activates the production of IL-1β and IL-18 via the NF-κB pathway. With the help of inflammasome complexes, CASP1 can be activated and cleaves precursor IL-1β and IL-18 [66].

As a significant member of NLR, NLRP3 is the main activator of inflammasomes after *H. pylori* infection. Koch et al. revealed that ROS production can activate the NLRP3 sensor after *H. pylori* invasion [67]. On the other hand, the expression of NLRP3 genes has been reported to be affected by *H. pylori* via the activated NOD2/NF-κB pathway [68]. For another inflammasome, NLRC4, Semper et al. showed that NLRC4 and downstream IL-18 can be activated by *H. pylori* through T4SS to cause gastritis and bacterial immune evasion [69]. The loss of NOD1 has been reported to increase the inflammatory responses and regulate the pro-inflammatory responses caused by *H. pylori* infection [70,71,72]. Interestingly, Castaño-Rodríguez et al. revealed that polymorphisms in the NLR signaling pathway increase the risk of the transformation to gastric cancer after *H. pylori* infection, which involves the regulation of NLRC4, NLRC5, NLRP9, NLRP12 and NLRX1 genes in *H. pylori*-challenged cells [73]. Inflammation in the stomach over time may lead to ulceration. The NLRs and NLR signaling pathways may also play an important role in the progression of *H. pylori*-associated gastritis. A study in 2023 on peptic ulcer disease found that the expression of NLRC4, NLRP12, IL-18 and IL-1β decreased significantly in patients of peptic ulcer disease compared with those of only *H. pylori*-associated gastritis [74].

### 3.2. IBD

Inflammatory bowel diseases (IBDs) are chronic inflammatory disorders of the gastrointestinal tract, including ulcerative colitis and Crohn’s disease. According to a report published in 2023, IBD impacts >0.7% of the population in the US [75]. In addition, the incidence of IBD peaks in early adulthood and then levels off at a lower rate [75]. The pathogenesis of IBD is complex and multiple factors—including host genetic predisposition, environmental exposure and microbial imbalance—may lead to mucosal inflammation and damage. Recently, cumulative studies have shown that the NLR family plays an essential role in the initiation and the process of IBD (Figure 2).

#### 3.2.1. Ulcerative Colitis

Ulcerative colitis is characterized by symptoms including abdominal pain, diarrhea and hematochezia with mucus. Although the specific pathogenic mechanism of ulcerative colitis is still unclear, the significant roles of the NLR family have gradually been revealed by recent research.

The signaling of NOD2 involved in ulcerative colitis can be intricate, which is a result of their extensive distribution in various intestinal epithelium and immune cells and complicated downstream pathways, such as the NF-κB and MAPK signaling pathways, as mentioned above. Under some circumstances, NOD2 can promote intestinal inflammation in ulcerative colitis. A study by Jamontt et al. found that colitis was enhanced by NOD2 signaling in IL-10-deficient mice. Also, after NOD2 was deleted in *IL-10*^−/−^ mice, the symptoms of colitis were significantly improved [76]. NOD2 was shown to be activated to stimulate the NF-κB and IF-17F pathways via CARD3 after *Fusobacterium nucleatum* (*F. nucleatum*) infection, which is associated with ulcerative colitis [77]. NOD2 participates in maintaining the regulation of intestinal intraepithelial lymphocytes (IELs). A study suggested that *Nod2*^−/−^ mice showed higher paracellular permeability and susceptibility to dextran sulfate sodium (DSS)-induced colitis [78]. Furthermore, a study suggested that NOD2 could change the microbial microenvironment to promote the risk of colitis because microbiota in NOD2-deficient mice increased the opportunity of colonic mucosa injury [79].

Another subfamily of NLR, NLRP, is also tightly connected with ulcerative colitis. Among NLRPs, the expression of NLRP3 has been demonstrated to be upregulated in ulcerative colitis, leading to tissue injury and clinical characterizations [80]. In 2019, a study by Chen et al. found that a significant component of the NLRP3 inflammasome called NEK7 can interact with NLRP3 to impact DSS-induced chronic colitis via pyroptosis after NF-κB activation [81]. However, NLRP3 was also found to act as a negative regulator in the initial immune response in ulcerative colitis. A series of research showed that mice deficient in NLRP3 signaling components, including NLRP3, ASC and CASP1, were more likely to develop DSS-induced colitis [82]. Also, the secretion of anti-inflammatory cytokines IL-10 and TGF-β can be associated with NLRP3 because they were downregulated in the colon of *Nlrp3*^−/−^ mice [83]. A study in 2017 found that the anomalously activated NLRP3 inflammasome caused the local increased production of IL-1β, induced regulatory T cells and maintained gut homeostasis through remodeling the gut microbiota to resist the colitis [84].

In addition, a study showed that a higher risk of developing ulcerative colitis was associated with carriers with a gene variant of NLRC4 [85]. As NLRC4 and NLRP3 have been demonstrated to be able to recruit the same inflammasome complex, the signaling of NLRC4 in ulcerative colitis may be similar to that of NLRP3. Recently, a large number of studies on other NLRP members involved in ulcerative colitis have emerged. NLRP1 gene expression has been found to be elevated in inflamed regions of ulcerative colitis patients’ colons. NLRP1 activity showed a positive correlation with IL-18 production and IFN-γ gene expression, which could exacerbate the inflammation response in ulcerative colitis [86]. NLRP12 gene expression was shown to have a negative correlation with active ulcerative colitis [87]. Furthermore, the p.S361L variant of the NLRP7 gene has been demonstrated to correlate with a significantly increased risk of ulcerative colitis [88]. Taken together, both the NLRP12 and NLRP7 signaling pathways can be important components of ulcerative colitis pathogenesis. Moreover, NLRX1 has been associated with the development of IBD. As a negative regulator of the NF-κB signaling pathway, NLRX1 can bind to IKKα or be associated with TRAF6 to form a TRAFasome complex, which disrupts downstream signaling and decreases IL-6 production [89]. The loss of NLRX1 in small intestinal epithelial cells (IECs) leads to higher sensitivity to DSS-induced colitis, which could be a result of a gut microbiome imbalance [90]. Leber et al. showed that NLRX1 regulated the adaptive response in IBD because a greater number of Th17 and Th1 cells with a higher proliferative capacity appeared in the colon of *Nlrx1*^−/−^ mice treated with DSS [91].

Interestingly, NLRs appear to link IBD disease severity with *H. pylori*-associated gastritis. A study in 2020 focusing on serum-derived exosomes from *H. pylori*-associated gastritis found that these exosomes upregulated NLRP12 in IECs to further inhibit the Notch signaling pathway. Subsequently, intestinal epithelial chemokines MCP-1 and MIP-1α were downregulated and the mice’s symptoms of DSS-induced colitis were lessened [92].

#### 3.2.2. Crohn’s Disease

Crohn’s disease is characterized by transmural lesions, including noncaseating granulomas being found in 60% of adolescent patients. Unlike ulcerative colitis, Crohn’s disease impacts the whole gastrointestinal tract, from mouth to anus. NOD2 gene polymorphism has been implicated in the etiology of Crohn’s disease through genome-wide association studies (GWAS) and meta-analyses [93]. Moreover, the risk of developing Crohn’s disease has been linked to genes which regulate components in the NOD2 signaling pathway, like ATG16L1, CARD9 and RIPK2 [94]. ATG16L1 is an autophagy protein on the alternative branch of the typical NOD2 signaling pathway. In a study, ATG16L1 was demonstrated to be a negative regulator of the NOD1/NOD2 signaling pathway by downregulating the activation of RIP2 [95]. As mentioned previously, RIPK2 facilitates the signal transduction between activated NOD2 and NF-κB. In addition, the polymorphism of NOD2 gene variants in Crohn’s disease led to decreased levels of NF-κB activation and an attenuated response to MDP-stimulation [96].

Other members in the NLR family also displayed a correlation with Crohn’s disease. NLRP3 inflammasome, as a major element in innate immunity, may protect against Crohn’s disease because a loss-of-function in NLRP3 was closely associated with the development of Crohn’s disease [97]. Furthermore, a study suggested that the occurrence of missense mutation in the CARD8 gene could activate NLRP3 inflammasome to affect Crohn’s disease [98]. Gao et al. reported that polymorphisms of IL-18 contributed to a higher susceptibility to Crohn’s disease [99]. A study in 2018 indicated that the expression of NLRP6 increased with Crohn’s disease activity [100].

The studies on NLRs’ involvement in IBD are summarized in Table 1.

### 3.3. Celiac Disease

Celiac disease is a chronic autoimmune disorder caused by a complex inflammatory response on the small intestine mucosa in response to gluten protein in genetically susceptible individuals. In the general population, the prevalence of celiac disease ranges between 0.5% and 2% [101]. Interestingly, a study found that the initial pathological adaptive immune response elicited by p31–43 peptide in gliadin involves NLRP3 [102]. In addition, other NLR members were also found to be involved in the development of celiac disease. According to a study on the role of small intestinal epithelial cells (IECs) in celiac disease, NLRP2/6/8, as inflammasome sensors, participate in an IFN-γ-circle in which NLRP2/6/8 receives signals from intestinal microbes and promotes the production of the downstream molecules IL-18 and IFN-γ. IFN-γ can further upregulate the genes associated with immune defense in celiac disease. IFN-γ production in celiac disease can enhance the activity of CD4 intraepithelial lymphocyte to promote the adaptive immune response [103]. A study carrying out transcriptomic analysis showed significantly downregulated NLRX1 signaling in celiac disease patients, suggesting a potential correlation between the NLRX1 signaling pathway and gluten sensitivity [104].

## 4. NLRs in GI Cancers

### 4.1. Esophageal Cancer

Esophageal cancer has a high morbidity and mortality rate, and patients diagnosed with esophageal cancer generally have a poor prognosis. In 2020, 604,100 people were diagnosed with esophageal cancer. Among all of the cases, 59.2% occurred in eastern Asia [105]. Histologically, esophageal carcinoma can be divided into two main groups: esophageal adenocarcinoma (EAC) and esophageal squamous cell carcinoma (ESCC). Overall, studies on esophageal cancer-related NLRs are limited. Most studies of squamous cell carcinoma focus on NLRP3. The NLRP3 inflammasome is upregulated in ESCC tissues and can promote the progression of ESCC [106]. As a key inducer of inflammation, NLRP3 plays an important role in regulating immune cells and reshaping the immune microenvironment. Esophageal exposures, such as bacterial infections and food sources, trigger inflammation and cancer. Microbial imbalance is believed to be one of the important causes of inflammation and cancer. Esophageal mucosa is susceptible to infection by pathogenic bacteria, which can induce inflammation and cancer. A recent study in vitro and in vivo has shown that NLRP3 enrichment is strongly associated with *F. nucleatum* infection and leads to the enrichment of myeloid-derived suppressor cells (MDSCs). In the case of *F. nucleatum* infection, the survival rate of patients is reduced, and high levels of MDSCs will also lead to drug resistance to chemotherapy [107]. This information provides a new consideration for preventing and prolonging the survival of ESCC patients; however, the specific mechanism still requires further studies. Another *in vitro* and *in vivo* study has shown that *F. nucleatum* also activates NF-κB through the NOD1/RIPK2 pathway, leading to tumor progression [108]. Other studies have found that nitrosamines mediate the activation of NLRP3 inflammasome in human esophageal epithelial cells (HTE-1A) through mtROS, and induce the pyroptosis of cells through the NLRP3/CASP1/GSDMD pathway [109]. However, the results of the *in vivo* experiments and the detailed mechanism of how mtROS mediates NLRP3 remain unknown.

### 4.2. Gastric Cancer

Gastric cancer remains one of the leading causes of cancer-related death. In 2020, 768,793 people worldwide died from stomach cancer [6]. *H. pylori* infection is considered one of the major risk factors for stomach cancer. Peptidoglycans from *H. pylori* are recognized by NOD1 and NOD2, which activate RIPK2 and induce the phosphorylation of ERK and p38, facilitating MAPK and NF-κB signal transduction [110]. However, the effect of NOD1 on the occurrence of cancer is considered to be two-sided. It facilitates the elimination of *H. pylori*, but long-term exposure to *H. pylori* can also lead to inflammation-mediated cancer [111]. NLRP3 also plays an important role in *H. pylori* infection. However, NLRC4 and NLRC6 are not considered necessary for the activation of the inflammation caused by *H. pylori* [67]. For NLRP3, a two-step activation is required. First, microbial products induce the enhanced expression of pro-forms of proinflammatory cytokines. Subsequently, pro-CASP1 is activated and catalyzes the processing of pro-IL-1β or pro-IL-18 into a mature form. Kim et al. reported that the secretion of IL-1β required co-interaction between TLR2, NOD2 and NLRP3 [112]. Their studies on mouse bone marrow-derived dendritic cells (BMDCs) showed that *H. pylori* cagPAI and CagL, but not CagA, enhanced IL-1β production in DCs. However, a recent study showed that *H. pylori* CagA upregulated the expression of NLRP3 inflammasome-related molecules in human gastric epithelial cells, and promoted the secretion of IL-1β and IL-18 [112,113]. ROS is also involved in the signaling of inflammatory body activation [114]. After the formation of NLRP3 inflammasome, it can induce the hydrolysis of pro-CASP1 to CASP1. Mature CASP1 processes the pro-IL-1β and pro-IL-18 induced by NF-κB and other factors into IL-1β and IL-18 [115,116]. IL-1β is considered an integral factor in the development of gastric cancer [117,118]. IL-18 promotes the immune escape of gastric cancer cells by upregulating programmed cell death 1 (PD1) in NK cells, downregulating CD70 in tumor cells and inhibiting the CASP8-mediated apoptosis in gastric cancer cells [119,120,121]. In *H. pylori* infection, NLRP3 activation is modulated by negative feedback. Recent studies have found that tripartite motif-containing 31 protein (TRIM31) attenuated NLRP3 inflammasome activation in *H. pylori*-associated gastritis by affecting ROS and autophagy [122].

### 4.3. Colorectal Cancer

In 2020, colorectal cancer (CRC) ranked third globally in incidence and second in mortality [123]. Environmental factors, rather than heritable genetic changes, are thought to be responsible for most cases of CRC [124], with the major predisposing factors including intestinal damage, oxidative stress and inflammation of the colon in the setting of inflammatory bowel disease (IBD) [125]. Evidence suggests that patients with Crohn’s disease or ulcerative colitis have a 60% higher incidence of CRC than the general population [126]. Chronic inflammation causes mutation in intestinal cells and disrupts intestinal homeostasis, promoting the transformation of intestinal cells into cancer cells.

In recent years, the importance of NLRs in the pathogenesis of CRC has been reported in a number of studies (Table 2).

It is reported that NLRC3, NLRP1b (a subfamily of NLRP1), NLRP3, NLRP6 and NLRP12 play an important role in preventing the occurrence of CRC [36,134,143,144,145]. However, the role of NLRs in CRC is more complex, especially NLRP3. This may also explain why most research on NLRs has focused on NLRP3.

Although previous studies have shown that NLRP3 has a protective effect on CRC, and *Nlrp3*^−/−^ mice showed an increase in colitis and CRC development induced by DSS and AOM/DSS [144], recent studies seem to support the promoting effect of NLRP3 on CRC more. It was reported that a high expression of NLRP3 is associated with poor survival and prognosis for CRC [146]. Other studies have shown that NLRP3 played an important role in promoting tumor metastasis. The activation of the NLRP3 inflammasome in macrophages secretes the pro-inflammatory cytokine IL-1β, which promotes the invasion and migration of CRC cells by regulating the epithelial-mesenchymal transition (EMT) [147,148,149,150]. Ovarian tumor deubiquitinase 6 A (OTUD6A) is one of the factors that promoted NLRP3 activation in macrophages. OTUD6A improved the stability of NLRP3 by selectively cutting the K48-linked polyubiquitin chains at K430 and K689 by binding to the NACHT domain of the NLRP3 inflammasome, resulting in elevated IL-1β levels [151]. Another study found that 5-hydroxytryptamine (5-HT) and 5-HT biosynthesis rate-limiting enzyme TPH1 were overexpressed in CRC cells. The upregulation of 5-HT production and secretion enhanced NLRP3 inflammasome activation in macrophages and activated NLRP3 intensifies 5-HT secretion by increasing IL-1β [152]. In addition, the activation of Ral, a member of the Ras subfamily, promoted the development of colitis-associated CRC by activating the NLRP3 inflammasome [153]. By inhibiting NLRP3-dependent IL-18, RAI16 maintained intestinal homeostasis and inhibited tumor development [154]. The study has shown that increased IL-18 production induced C-X-C Motif Chemokine Ligand 16 (CXCL16) secretion, which recruited immunosuppressive bone MDSCs and enhanced tumor cell proliferation and migration [154]. However, it is important to note that the effect of IL-18 on tumors also has two sides [155]. The role of pro-inflammatory cytokines in tumor development is complex, which may be related to the different types and stages of cancers, as well as the complicated tumor microenvironment. The Akt/MTOR/S6K1 signaling pathway is involved in cell proliferation and malignant tumor development. The study suggested that NLRP3 may contribute to the development of CRC by interconnecting with MTOR-S6K1 via an intermediate MAPK signaling pathway [156].

The data showed that CRC incidence and mortality are about 25 percent higher in men than in women [157]. Therefore, attention should also be paid to the development of CRC caused by sex hormones. Son et al. found that estradiol inhibited CRC by downregulating the pro-inflammatory mediators related to NF-κB, promoting the activation of Nrf2-related antioxidant enzymes and the NLRP3 inflammasome. Estradiol significantly reduced AOM/DSS-induced inflammation in male mice [158]. It is worth noting that in this study, the dual role of Nrf2 and the NLRP3 inflammasome in tumors was observed; that is, they could prevent tumor development, but once the tumor develops, they have a promoting effect [158]. This phenomenon may partially explain the dual role of NLRP3 found in previous studies. In addition, Fan et al. identified, for the first time, the estrogen receptor (ER) as one of the transcriptional regulators of NLRs. ER participates in NLRs’ signal transduction pathways in CRC by directly regulating NLRs [159]. Further research found that ER could regulate the Wnt/β-catenin signaling pathway in cancer by targeting NLRs [160].

The intestinal microbiota is closely related to the occurrence of CRC, and the intestinal microbiota of CRC patients is often different from that of healthy people [161]. NLRs’ role in the development of CRC may be influenced by communication between the microbiome and the host immune system. However, the specific mechanism by which microorganisms affect tumor development through NLRs remains unclear. Recently, the mechanism of NLRP3 activation induced by *Candida tropicalis* in MDSCs has been discovered. The JAK-STAT1 signaling pathway acted as the first initiating signal to promote the transcription of the *NLRP3*, *pro-CASP1* and *IL-1β* genes. Then, the mtROS acted as the second activating signal to mediate the activation of NLRP3. The activation of NLRP3 promoted CRC progression [162]. Wang et al. found that *Porphyromonas gingivalis* (*P. gingivalis*) was also associated with the poor prognosis of CRC because *P. gingivalis* promoted the hematopoietic NLRP3 inflammasome [163]. In addition, statistical analyses suggested that *Hemophilus influenzae* may be associated with a reduced incidence of CRC associated with NLRP3; however, the exact mechanism remains unclear [164].

## 5. Therapies Targeting the NLR Signaling Pathway

### 5.1. Treatment Strategies in Inflammatory Bowel Disease

Accompanied by the ongoing research on the NLR signaling in GI inflammatory diseases, NLRs are regarded as potential targets to treat GI inflammatory diseases. Among all of the GI inflammatory diseases mentioned above, experiment agents targeting NLRs in IBD have been subject to the most studies (Table 3). In 2004, an early study showed that the p38 mitogen-activated protein kinase (MAPK) inhibitor SB203580 could suppress DSS-induced experimental colitis via inhibiting NOD2 effector RIPK2 in the NOD2/RIPK2/NF-κB signaling pathway [165]. Based on this idea, a selective 4-aminoquinoline-based RIP2 inhibitor, GSK583, has shown effectiveness in the intestinal mucosa of IBD patients by blocking NOD2 signaling [166].

NLRP3, another well-characterized NLR in IBD, has been an essential target in IBD treatment. Curcumin is a possible choice for IBD treatment. As a regulator of the NF-κB/IκB pathway, curcumin blocks the activation of NF-κB and further interrupts the following activation of the NLRP3 inflammasome. According to a study in 2018, curcumin could alleviate DSS-induced colitis in mice via the inhibition of the NLRP3 inflammasome with a reduction in IL-1β and IL-6. In addition, this study also showed that the other mechanisms triggering NLRP3 inflammasome activation, like ROS generation, K^+^ efflux and cathepsin B leakage, could be suppressed in DSS-induced colitis after curcumin treatment [167]. A large number of flavonoids have also shown potential effects on IBD treatment. Apigenin (API) is a flavone (4,5,7-trihydroxyflavone) that is ubiquitously distributed in plants and has shown the ability to prevent DSS-induced colitis via regulating CASP1’s activity to inhibit the activation of the NLRP3 inflammasome in the colon [168]. Similarly, other natural products in plants, like alpinetin, wogonoside and naringin, are potential agents of IBD treatment with the parallel mechanism [169,170,171]. In addition, a study demonstrated that another flavonoid, cardamonin, could alleviate IBD via upregulating the Nrf2/NQO1 pathway and further inhibiting NLRP3 inflammasome activation [172]. In addition, a small molecule inhibitor of NLRP3, MCC950, showed potential in preventing the activation of the NLRP3 inflammasome and effectiveness in a colitis mouse model.

Additional targets in the NLR signaling pathways are being investigated for GI inflammatory diseases. For instance, API regulates the NLRP6 signaling pathway to modulate the gut microbiota in DSS-induced colitis [173]. Secoisolariciresinol diglucoside (SDG), an inhibitor of the NLRP1 inflammasome, disrupts downstream NF-κB activation, which also ameliorated DSS-induced colitis [174]. At present, NX-13, a small molecule that activates NLRX1 signaling to reduce the overexpression of proinflammatory cytokines, was well-tolerated in a phase Ib clinical trial for IBD treatment [175]. A phase II clinical trial of NX-13 will be carried out. However, the therapeutic effects of the potential agents for IBD targeting NLR signaling were mostly carried out in mice. As a result, in the future, more attention should be paid to the clinical trials of these potential drugs and further elucidating the detailed molecular mechanisms of the associated NLR signaling pathways.

### 5.2. Treatment Strategies in Colorectal Cancer

NLRs are important players in the maintenance of gastrointestinal homeostasis, as well as affecting cancers associated with gastrointestinal inflammation; therefore, new therapies targeting NLRs for colorectal cancer may emerge (Table 3). Chemotherapy agents, such as 5-fluorouracil (5-FU)-based chemotherapy, are the first-line treatment for CRC. However, 5-FU activates the NLRP3 inflammasome in MDSCs, leading to the production of IL-1β. IL-1β induces the secretion of IL-17 by CD4^+^ T cells, resulting in curtailed anti-tumor immunity [176]. It has been reported that androstenimide sulfonate can reverse this process and restore cell sensitivity to 5-FU by inhibiting NLRP3, followed by an increase in IL-17 produced by CD4^+^ T cells and a reduction in IL-1β [177]. In addition, docosahexaenoic acid has been reported to improve 5-FU chemotherapy by inhibiting NLRP3 assembly and JNK-mediated IL-1β secretion [178]. Quercetin, the major plant flavonoid widely found in plants, shows low cytotoxicity and enhanced CRC cell inhibition after fermentation by gastrointestinal microorganisms. Fermented quercetin with *Lactobacillus plantarum* also improves cell sensitivity to 5-FU by inhibiting NLRP3 and phosphorylating ERK [179].

Curcumin has shown potential for the treatment of IBD [180,181]. Studies have also shown that curcumin could enhance the sensitivity of tumor cells to 5-FU [182]. Curcumin’s role in enhancing chemosensitivity to 5-FU may be associated with its role in inhibiting the NLRP3 inflammasome [177,183]. In addition, although the synergistic therapy of curcumin and 5-FU shows a broad application prospect, more studies and clinical trials are still needed to clarify the mechanism, efficacy and toxicity of the combination therapy.

Probiotics have shown great potential in regulating the balance of gut microbes and inhibiting the development of CRC [184]. *In vivo* and *in vitro* experiments showed that *Akkermansia muciniphila* promoted the enrichment of M1-like macrophages through TLR2/NLRP3-dependent signal transduction and regulated intestinal immunity, thereby inhibiting CRC [185]. Heat-killed *Enterococcus faecalis* (*E. faecali*) inhibited the activation of the NLRP3 inflammasome in THP-1-derived macrophages. A mouse model of CRC induced by AOM/DSS showed that *E. faecalis* improved CRC symptoms [186]. Bacteroides fragilis inhibited CRC by producing butyrate to inhibit NLRP3 in macrophages to reduce the levels of IL-1β and IL-18 [183]. Notably, another study found that *Inonotus obliquus* polysaccharide (IOP), the primary constituent of the parasitic fungus *Inonotus obliquus*, activated the NLRP3 inflammasome in CRC cells, and IL-1β and IL-18 were upregulated rather than downregulated to exert the inhibitory effect of CRC [187]. Gao et al. recently reported that a bi-functional peptidoglycan hydrolase present in most probiotics was capable of producing muramyl dipeptide (MDP), a NOD2 ligand, to protect inflammation-associated CRC through MDP-NOD2 signaling [188]. Overall, NLRs are important targets for the treatment of CRC.

**Table 3 ijms-24-14511-t003:** Novel agents targeting NLRs for IBD and colorectal cancer treatment.

Targets Related to NLR Signaling Pathways	Disease	Compounds	Authors	Findings	Reference
NOD2	IBD ^1^	SB203580	Hollenbach et al.	Inhibiting NOD2/RIPK2/NF-κB signaling pathway in DSS-induced colitis	[165]
	IBD	GSK583	Haile et al.	Inhibiting NOD2 signaling in IBD patients	[166]
NLRP3	IBD	Curcumin	Gong et al.	Inhibiting the activation of NF-κB and NLRP3 inflammasome in DSS-induced colitis	[167]
	IBD	Apigenin	Márquez-Flores et al.	Inhibiting the activation of CASP1 and NLRP3 inflammasome in DSS-induced colitis	[168]
	IBD	Alpinetin	He et al.	Inhibiting the activation of NF-κB and NLRP3 inflammasome in DSS-induced colitis	[169]
	IBD	Wogonoside	Sun et al.	Inhibiting the activation of NF-κB and NLRP3 inflammasome in DSS-induced colitis	[171]
	IBD	Naringin	Cao et al.	Inhibiting the activation of MAPK and NLRP3 inflammasome in DSS-induced ulcerative colitis	[170]
	IBD	Cardamonin	Wang et al.	Inhibiting Nrf2/NQO1 signals and NLRP3 inflammasome in in DSS-induced colitis	[172]
	IBD	MCC950	Wang et al.	Inhibiting the activation of NLRP3 inflammasome in DSS-induced colitis	[189]
	CRC ^2^	Galloflavin	Guo et al.	Reducing NLRP3 expression and inflammatory factors level, while reducing the expression of c-Myc and p21	[190]
	CRC	Arctigenin	Qiao et al.	Downregulating fatty acid oxidation to inhibit NLRP3 inflammasome assembly in macrophages, which leads to a decrease in IL-1β	[191]
	CRC	Caffeic acid phenethyl ester	Dai et al.	Inhibited CRC by inhibiting the production of reactive oxygen species to promote the ubiquitination of NLRP3	[192]
	CRC	Docosahexaenoic acid	Dumont et al.	Inhibiting NLRP3 assembly and JNK-mediated IL-1β secretion to improve 5-FU chemotherapy	[178]
	CRC	Andrographolide sulfonate	Xu et al.	Inhibiting the activation of NLRP3 inflammasome in myeloid-derived suppressor cells, followed by the increase in IL-17 produced by CD4^+^ T cells and reduction in IL-1β to sensitize 5-FU treatment	[177]
	CRC	FL118	Tang et al.	Inhibiting the growth and metastasis of CRC by inducing NLRP3-ASC-CASP1 mediated pyroptosis	[193]
	CRC	Atractylenolide I	Qin et al.	Inhibiting NLRP3 inflammasome activation by suppressing Dynamin-related protein 1-mediated mitochondrial fission	[194]
	CRC	Fermented quercetin	Lee et al.	Downregulating expression of NLRP3 and phosphorylation of ERK to improve cell sensitivity to 5-FU	[179]
	CRC	Ginsenoside Rh3	Wu et al.	Triggering pyroptotic cell death and ferroptotic cell death in CRC cells via the Stat3/p53/Nrf2 axis	[195]
	CRC	Huoxiang Zhengqi	Dong et al.	Regulating the composition of intestinal microbiome and metabolism; activated Nrf2 mediated antioxidant response and inhibited NF-κB mediated NLRP3 activation	[196]
	CRC	Oxymatrine	Liang et al.	Reducing mitophagy-activated NLRP3 inflammasome through LRPPPRC inhibition	[197]
NLRP6	IBD	Apigenin	Radulovic et al.	Regulating NLRP6 signaling pathway in DSS-induced colitis	[173]
NLRP1	IBD	Secoisolariciresinol diglucoside	Wang et al.	Inhibiting NLRP1 inflammasome in DSS-induced colitis	[174]
NLRX1	IBD	NX-13	Leber et al.	Activating NLRX1 signaling in DSS-induced colitis	[175]
NLRC3	CRC	Dihydromethysticin	Pan et al.	Affecting cell proliferation, migration, invasion, apoptosis, cell cycle and angiogenesis via NLRC3/PI3K pathway to inhibit CRC	[198]
	CRC	The Green Walnut Husks	Chen et al.	NLRC3/PI3K/AKT pathway to regulate the levels of mTOR, Bcl-2 and Bax to promote apoptosis of tumor cells and inhibit cell proliferation, invasion and migration to prevent CRC progression	[199]

^1^ IBD, inflammatory bowel diseases; ^2^ CRC, colorectal cancer.

## 6. Conclusions and Future Perspectives

In recent years, there has been growing evidence that NLRs play a vital role in GI inflammatory diseases and GI cancers. NLRs, as intracellular sensors of PAMPs and DAMPs, play an important role in activating the host responses to pathogen infection and cellular stress. NLRs can also be used as biomarkers and potential treatment targets for GI diseases. In this review, we summarized the existing research results on the roles and mechanisms of NLRs in GI inflammatory diseases and GI cancers. We also reviewed the potential therapies targeting NLRs in several of these diseases.

A significant number of the published studies on NLR signaling in GI diseases focus on the typical NLR signaling pathways, including NOD2 and NLRP3. Additional future studies will provide a more complete picture of the intricate relationship between the various components of this complex immune regulatory system, including the role of downstream cytokines such as IL-17F, as well the involvement of NLRs in adaptive immunity, such as in the case of IBD.

As many of the GI inflammatory diseases and GI cancers are influenced by host genetic predisposition and environmental factors such as diet, infections and microbiome, a more thorough study of how NLRs play a role in this process will be needed. Some evidence has emerged that genetic polymorphisms in the NLR signaling pathway genes influenced disease risk and severity. However, our understanding in this area is still quite limited, and it will be a promising area for future research.

Animal model studies of the relationship between NLR signaling and disease development require better standardization in terms of the experimental conditions, including the sex of the animal, animal housing conditions and duration of treatment with AOM/DSS. By analyzing ten CRC databases on the Oncomine^®^ platform by Liu et al., it is shown that the consistency of the NLRP3 expression levels was poor across different databases, which might lead to difficulty in interpreting different results from various studies [200]. In addition, although animal model studies have shown great promise in treatment strategies targeting NLRs, future well-planned human clinical trials are needed to move novel and promising therapies to the clinic.

## Figures and Tables

**Figure 1 ijms-24-14511-f001:**
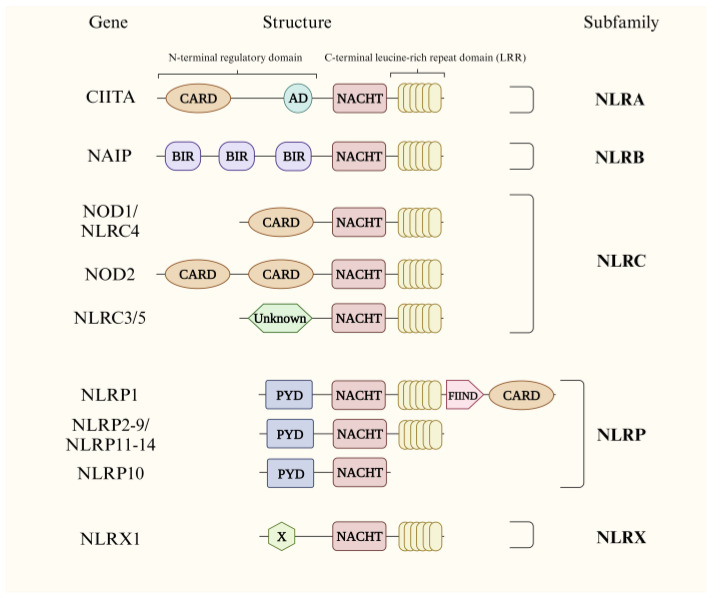
Schematic representation of the protein structure of each member in NLR family. NLRs in humans are divided into five groups, including NLRA, NLRB, NLRC, NLRP and NLRX. The typical structure of NLR proteins consists of an N-terminal regulatory domain, a NACHT domain and a C-terminal leucine-rich repeat domain (LRR). AD, acidic transactivation domain; BIR, baculovirus inhibitor repeat domain; CARD, caspase recruitment domain; CIITA, class II major histocompatibility complex transactivator; FIIND, function to find domain; LRR, leucine-rich repeat domain; NACHT, nucleotide-oligomerization-binding domain; NAIP, neuronal apoptosis inhibitor protein; PYD, pyrin domain; X, unidentified.

**Figure 2 ijms-24-14511-f002:**
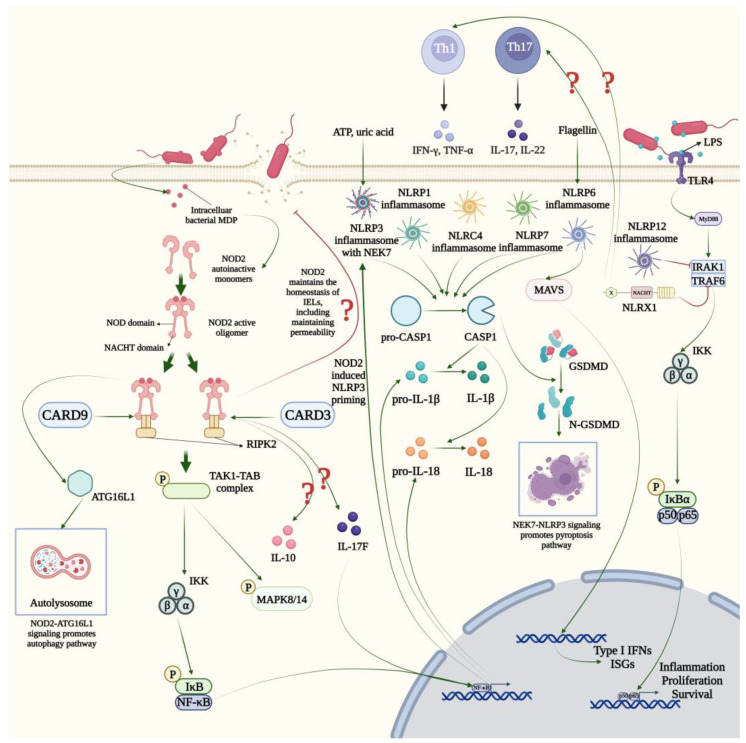
Major and other possible molecular pathways associated with NLRs in IBD. NOD2 recognize the bacterial peptidoglycan MDP. NOD2 monomers are oligomerized by CARD-CARD interactions to recruit RIPK2, which further activates NF-κB or MAPK signaling pathway. Meanwhile, the NOD2 signalosome promotes the production of cytokines and activates autophagy pathway via ATG16L1. Both the NOD2-induced priming and environment factors like ATP activate the NLRP3 inflammasome. The maturation of CASP1 is induced by inflammasomes of NLRP1, NLRP3, NLRP6, NLRP7 and NLRC4, which assists the production of IL-1β and IL-18. With the help of NEK7, NLRP3 inflammasome further promotes the pyroptosis pathway. In addition, NLRP12 inflammasome and NLRX1 inhibit the TLR4 signaling pathway which is associated with the production of interferons. ATG16L1, autophagy-related 16-like 1 protein; CARD3, caspase activation and recruitment domain 3 protein; CARD9, caspase activation and recruitment domain 9 protein; CASP1, caspase 1; GSDMD, gasdermin; IFN-γ, interferon-gamma; IκB, I kappa B protein; IKK, I kappa B kinase; IL, interleukin; IRAK1, interleukin-1 receptor-associated kinase 1; ISGs, interferon-stimulated genes; LPS, lipopolysaccharide; MAPK8/14, mitogen activated protein kinase 8/14; MAVS, mitochondrial antiviral signaling protein; MDP, muramyl dipeptide; MyD88, myeloid differentiation factor 88; NEK7, NIMA (never in mitosis gene a)-related kinase 7; NF-κB, nuclear factor-kappa B; p50/p65, NF-κB p50/p65 subunit; RIPK2, receptor-interacting serine-threonine kinase 2 protein; TAB, TAK1-associated binding protein; TAK1, transforming growth factor-beta-activated kinase 1; Th1/17, type 1/17 helper T cell; TLR4, toll like receptor 4; TNF-α, tumor necrosis factor alpha; TRAF6, TNF receptor associated factor 6; X, unidentified.

**Table 1 ijms-24-14511-t001:** Summary of NLRs’ roles in IBD.

NLR Signaling Pathways	Disease	Authors	Findings	Reference
NOD2	UC ^1^	Jamontt et al.	NOD2 signaling in IL-10-deficient mice enhanced ulcerative colitis	[76]
	UC	Chen et al.	NOD2 activated NF-κB and IF-17F pathways via CARD3 after *Fusobacterium nucleatum* infection associated with ulcerative colitis	[77]
	UC	Natividad et al.	NOD2 was associated with paracellular permeability and susceptibility in DSS-induced colitis	[78]
	UC	Couturier-Maillard et al.	NOD2 changed change the microbial microenvironment to promote the risk of ulcerative colitis	[79]
	CD ^2^	Jostins et al.	NOD2 gene polymorphism has been implicated in the etiology of Crohn’s disease	[93]
	CD	Franke et al.	The risk of developing Crohn’s disease has been linked to genes regulating components in the NOD2 signaling pathway	[94]
	CD	Ashton et al.	Polymorphism of NOD2 gene variants in Crohn’s disease led to decreased levels of NF-κB activation	[96]
NLRP3	UC	Ranson et al.	The expression of NLRP3 was upregulated in ulcerative colitis	[80]
	UC	Chen et al.	NEK7 interacted with NLRP3 to impact DSS-induced colitis via pyroptosis	[81]
	UC	Hirota et al.	Anti-inflammatory cytokines IL-10 and TGF-β decreased in the colon of *Nlrp3*^−/−^ mice	[83]
	UC	Yao et al.	Anomalously activated NLRP3 inflammasome remodeled the gut microbiota to resist the colitis	[84]
	CD	Zhen et al.	Loss-of-function in NLRP3 was closely associated with the development of Crohn’s disease	[97]
	CD	Mao et al.	Mutated CARD8 gene activated NLRP3 inflammasome to affect Crohn’s disease	[98]
NLRC4	UC	Steiner et al.	A gene variant of NLRC4, NLRC4 (A160T), showed a higher risk of developing ulcerative colitis	[85]
NLRP1	UC	Tye et al.	NLRP1 upregulated IL-18 and IFN-γ in ulcerative colitis	[86]
NLRP12	UC	Chen et al.	NLRP12 gene expression diminished in active ulcerative colitis	[87]
NLRP7	UC	Onoufriadis et al.	A gene variant of NLRP7, NLRP7 (p.S361L), showed a higher risk of developing ulcerative colitis	[88]
NLRP6	CD	Ranson et al.	The expression of NLRP6 increased with Crohn’s disease activity	[100]
NLRX1	UC	Leber et al.	Loss of NLRX1 led to higher sensitivity to DSS-induced colitis	[90]
	UC	Leber et al.	A greater number of Th17 and Th1 cells with higher proliferative capacity appeared in the colon of DSS-treated *Nlrx1*^−/−^ mice	[91]

^1^ UC, ulcerative colitis; ^2^ CD, Crohn’s disease.

**Table 2 ijms-24-14511-t002:** Summary of studies on NLRs and colorectal cancer.

NLR Signaling Pathways	Authors	Findings	Reference
NLRC3	Karki et al.	NLRC3 acted as an inhibitory sensor of PI3K-mTOR, mediating protection against colorectal cancer	[36]
	Karki et al.	NLRC3 attenuated the development of colorectal cancer by suppressing c-Myc expression and activation of PI3K-AKT targets FoxO3a and FoxO1, which regulate cellular proliferation	[127]
NLRC4	Hu et al.	CASP1, mediated by the NLRC4 inflammasome, prevented colon inflammation-induced tumors by regulating the proliferation and apoptosis of colon epithelial cells	[128]
NLRP3	Liu et al.	Deoxycholic acid disrupted the intestinal mucosa barrier and induced intestinal low-grade inflammation, which promoted intestinal tumorigenesis	[129]
	Cambui et al.	rs2072443 variant in *TMEM176B* improved the prognosis of colorectal cancer	[130]
	Tezcan et al.	Rab5 enhanced the activation of NLRP3, while Rab7 and Rab11 played a role in enhancing the expression of NLRP3 gene	[131]
NLRP6	Frühbeck et al.	Downregulation of NLRP6 and IL-18 in the colon of colon cancer patients in the context of obesity led to reduced intestinal barrier integrity, triggering a vicious cycle of inflammatory cascades under the action of adipose tissue	[132]
NLRP7	Li et al.	NLRP7 deubiquitination by USP10 promoted tumor progression and tumor-associated macrophage polarization in colorectal cancer	[133]
NLRP12	Zaki et al.	NLRP12 negatively regulated NF-κB and ERK signaling in macrophages to inhibit pro-inflammatory cytokines and chemokines, thereby inhibiting colorectal tumorigenesis	[134]
	Allen et al.	NLPR12 acted as a negative regulator of non-canonical NF-κB pathway, accompanied by activation of ERK and induction of NIK-dependent gene	[46]
	Kanneganti et al.	NLRP12 involved in the regulation of the Wnt/β-catenin pathway, inhibiting colorectal tumors by promoting β-catenin degradation	[135]
NLRX1	Tattoli et al.	NLRX1 played a role in colorectal cancer prevention by inhibiting TNF-induced intestinal epithelial cell proliferation	[136]
	Koblansky et al.	NLRX1 acted as a tumor suppressor by attenuating *Apc^min^*^/+^ colon tumorigenesis, cellular proliferation, NF-κB, MAPK, STAT3 activation and IL-6 levels	[137]
NOD1	Zhan et al.	NOD1 limited colitis-associated tumorigenesis by regulating IFN-γ production	[138]
	Jiang et al.	NOD1 increased colon cancer cell adhesion, migration and metastasis through the p38 MAPK pathway	[139]
	Maisonneuve et al.	Myeloid-intrinsic NOD1 expression maintained intra-tumor arginase-1 levels to foster an immunosuppressor and tumor permissive microenvironment that promotes tumor development	[140]
	Wei et al.	CDC42 promoted liver metastasis of colorectal cancer cells by activating NOD1 in macrophages	[141]
NOD2	Udden et al.	NOD2 inhibited colorectal cancer by downregulating TLR-mediated NF-κB and MAPK pathways	[142]

## Data Availability

Not applicable.

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
