# Peer review of "NOD-like Receptor Signaling Pathway in Gastrointestinal Inflammatory Diseases and Cancers"

_ijms, 2023, doi:10.3390/ijms241914511_

Round 1
Reviewer 1 Report
Manuscript: 2599822
In this review manuscript titled " NOD-like receptor signaling pathway in gastrointestinal inflammatory diseases and cancers " by Zhou et al., the authors aimed to gather most of the literature available on NOD-like receptor involvement in gastrointestinal inflammatory disease and cancers. Though authors tried to draft a comprehensive review on NOD-like receptors but the manuscript lacks originality or novelty.
Most of the text and figures constructed in the manuscript has already been reviewed and published. Please refer;
“NOD-like receptor signaling in inflammation-associated cancers: From functions to targeted therapies” by Liu et al (PMID: 31465982) and
“NOD-Like Receptors: Master Regulators of Inflammation and Cancer” by Saxena and Yeretssian (PMID: 25071785)
I recommend authors to provide information that are not already been extensively reviewed. Take out all the information that has been published by citing the respective articles and layout only those that are absolutely necessary. Overall, this review has to be restructured and outlined in a different approach than the available reviews.
English is fine, however, proof reading is required. A few sentence and grammar errors are noted.
Author Response
Dear Reviewer 1:
Thanks for your comments! For our detailed point-by-point response, please see the attachment.

Reviewer 2 Report
Topic of manuscript is interesting and suitable of the IJMS. Nevertheless, to increase its clarity, readability and impact some point can be given.
Figure 1-functionality of protein domains and types should be illustrated. If this is not possible use the tables for this purpose
In the subchapter chapter 5.1 You can mention curcumin (inhibitor NF-Kb signalling).
In the subchapter chapter 5.2 Possible synergy of flavonoids and curcuminoids with mentioned drugs should be also discussed.
Table/s for the summarization of biological and clinical studies should be includes into manuscript.
Subchapter “Future perspectives” is too short. Future development (newly discovered hypotheses, therapeutic application and limitations and their possible overcoming) should be discussed based on the relevant studies.
Minor
Paragraphs are sometimes to long for the comfortable orientation in the text.
Author Response
Dear Reviewer 2:
Thanks for your comments! For our detailed point-by-point response, please see the attachment.

Reviewer 3 Report
The review “NOD-like receptor signaling pathway in gastrointestinal inflammatory diseases and cancers” is an interesting and important subject selected by the authors to compile updated information related to the topic.
The authors have made a good attempt and have written a comprehensive review of the topic. English grammar and sentence framing need to be improved in the whole manuscript.
Comments:
1. Some sentences are lengthy and confusing. Example, lines 27-31. Authors are requested to shorten and simplify the sentences in the whole article where it is present.
2. In the introduction section, the authors talk about GI and cancer. It is not clear whether the authors are focusing only on gastrointestinal cancers or cancers in general. Authors should very clearly state what exactly they are talking about.
3. Authors should provide updates on the statistics of GI incidences (different types of GI, incidences in males and females, etc.,) worldwide and also of gastrointestinal cancers. A table of that will give a good impression to the readers.
4. Lines 59-61 look inappropriate, reframe it.
5. Lines 74-78 can be written after line 66.
6. In section 3.1., authors should add a couple of sentences on what gastritis is before starting with H. pylori associated gastritis.
7. Line 207-208, “Gastric inflammatory responses are largely induced by the secretion of virulence factors from H. pylori through T4SS”, Authors should expand/elaborate on what is T4SS. Likewise, in other places also, authors should elaborate before using abbreviations.
8. Line 212, Don’t use the term Hp. Authors should keep uniformity in using the names or any terms in the whole manuscript.
9. A short table can be made for the therapeutic strategies for GI inflammatory diseases and GI cancers which can include the name of the compound, target receptor type, disease in which it is used, reference, etc.
10. The starting sentence of the conclusion is not well written. Reframe it. Also, in the last sentence, instead of using “will be” authors should use “should be”.
The English language needs extensive improvement in the whole manuscript.
Author Response
Dear Reviewer 3:
Thanks for your comments! For our detailed point-by-point response, please see the attachment.

Round 2
Reviewer 1 Report
Authors have addressed the concerns and provided clarification.
Reviewer 2 Report
I have no objection.
Reviewer 3 Report
The authors have made significant changes to the manuscript as per the suggestions. It is now in good shape and suitable for publication.
Authors need to improve the English language and check some of the grammatical errors in the manuscript. It will improve the overall quality of the manuscript.
The English language needs to be refined.